# Infantile Hemangiomas of the Head and Neck: A Single-Center Experience

**DOI:** 10.3390/children11030311

**Published:** 2024-03-06

**Authors:** Deniz Kizmazoglu, Dilek Ince, Yuksel Olgun, Emre Cecen, Handan Guleryuz, Taner Erdag, Nur Olgun

**Affiliations:** 1Department of Pediatric Oncology, Institute of Oncology, Dokuz Eylul University, Izmir 35340, Turkeyemre.cecen@deu.edu.tr (E.C.); nur.olgun@deu.edu.tr (N.O.); 2Department of Otorhinolaryngology, Faculty of Medicine, Dokuz Eylul University, Izmir 35340, Turkeytaner.erdag@deu.edu.tr (T.E.); 3Department of Radiology, Faculty of Medicine, Dokuz Eylul University, Izmir 35340, Turkey; handan.guleryuz@deu.edu.tr

**Keywords:** infantile hemangioma, head and neck, treatment, propranolol, complications

## Abstract

**Background:** Infantile hemangiomas (IHs) are the most common benign vascular tumors of infancy. **Methods:** We report our experiences with 248 patients with head and neck IHs. **Results:** The median admission age was 4 months, and the female/male ratio was 2.18. Among the cases, 45% were followed by no treatment. No local complications were observed in any of these patients. Propranolol was provided to all patients who received medical treatment. The median duration of treatment was 12 months (1–30 months), and the median follow-up period of all patients was 14 months (0–118 months). The treatment response was 98%. The complication rate was 17%, and children aged between 3 and 9 months accounted for 60% of the patients who developed complications. Most of the complications were local complications, such as ulceration and bleeding. **Conclusions:** Although most IHs regress spontaneously, complications may occur. Propranolol alone is an effective treatment option, and early treatment initiation increases the success rate.

## 1. Background

Infantile hemangiomas (IHs) are the most common benign pediatric vascular tumors, characterized by abnormal proliferation of endothelial cells. The incidence varies between 1 and 3% in newborns and 3 and 10% in infants under 12 months of age. The prevalence is significantly higher in female, preterm, and low-birth-weight infants. Family history is not suggested as a risk factor. The majority of IHs are not present at birth. Precursor lesions (pale area of vasoconstriction, an erythematous macule, a telangiectatic red macule, or blue bruise-like patches) can sometimes be observed from birth. Clinically, a proliferative phase and a plateau and involution phase can be observed. After the rapid postnatal growth of the lesion, growth stops, and the lesion spontaneously regresses and heals, leaving a burn-like scar (fatty fibrous residue). The pathogenesis of IHs remains unknown. Angiogenic and vasculogenic factors, and tissue hypoxia are thought to contribute to the development of IHs. GLUT-1 is the immunochemical marker of infantile hemangiomas [1,2].

The majority of IHs regress spontaneously and require no treatment. The need for medical treatment is determined by the location of the IH, complications, organ dysfunction caused by the lesion, and functional losses. A surgical approach is not a priority in IHs that are expected to regress spontaneously or that can be controlled with medical treatment. Surgery is sometimes performed for reconstruction after the fatty fibrous residue phase [1,3].

Vascular malformations are important in the differential diagnosis of IHs. Mulliken and Glowacki [4] first classified vascular anomalies in 1982. In 1996, this classification was adopted and expanded by the International Society for the Study of Vascular Anomalies (ISSVA), which was modified and last updated in 2018 [5,6,7,8]. Infantile hemangioma is the most common benign vascular tumor, and other benign vascular tumors include congenital hemangioma, tufted hemangioma, spindle-cell hemangioma, epithelioid hemangioma, and pyogenic granuloma, amongst others. Congenital hemangiomas, which are present at birth, have two variants: non-involuting congenital hemangioma (NICH) and rapidly involuting congenital hemangioma (RICH). Vascular malformations are congenital anomalies, often present at birth, and the rate of endothelial cells is within the normal range. Vascular malformations expand as the patient grows, and do not regress spontaneously [1].

Infantile hemangiomas are mainly classified as focal, multifocal, or segmental, according to their distribution. Around 80% of infantile hemangiomas arise from a single point and are focal. The least common type, segmental hemangiomas, involve a large area. Segmental hemangiomas of the head and neck are more commonly involved in *PHACE* syndrome (*P*osterior fossa defects, *H*emangiomas, cerebrovascular *A*rterial anomalies, *C*ardiovascular anomalies including coarctation of the aorta, and *E*ye anomalies). *PHACE* syndrome requires a more aggressive and multidisciplinary approach. They are also classified as superficial, deep, or mixed according to their localization. Superficial hemangiomas appear earlier than deep hemangiomas, and the involution phase begins sooner [1,2,8]. Most IHs are mainly diagnosed based on physical examination and clinical history. Radiological evaluation is occasionally needed to differentiate deep IHs from vascular malformations or soft tissue tumors [1]. Approximately 60% of IHs are located in the head and neck region, followed by the trunk, extremities, and genitalia. Infantile hemangiomas are rarely located in the muscle, bone, liver, spleen, lymph node, thymus, gastrointestinal system, lung, salivary gland, brain, and spinal cord. Most head and neck hemangiomas are found around the orbit, mouth, and the nose [9].

Observation is the main treatment approach to IHs. Medical treatment is indicated when vital functions are affected. Function-threatening complications require treatment decision. Ulceration is the most common complication, followed by respiratory distress, feeding difficulties, bleeding, visual impairment, infections, and auditory impairment. Ulceration can lead to bleeding, pain, and infection [1]. Propranolol is the first-line treatment for IHs. Propranolol is a lipophilic, nonselective β-blocker. It is traditionally used for cardiovascular disorders. Its mechanism of action in hemangiomas can be summarized as vasoconstriction and the inhibition of apoptosis and angiogenesis. After the initiation of propranolol therapy, an improvement has been noted within at least three months [1,2,9,10].

In this study, we aimed to evaluate the clinical features of patients diagnosed with head and neck IHs and to assess their treatment indications, complications, and treatment responses.

## 2. Materials and Methods

The records of a total of 410 patients admitted with a diagnosis of IH to Dokuz Eylül University, Oncology Institute, Department of Pediatric Oncology, between 1 January 2010 and 1 January 2022, were followed and retrospectively analyzed. Patients presenting with head and neck IHs were selected for this study. All patients with head and neck IHs were diagnosed by a multidisciplinary team including a pediatric oncologist, an otolaryngologist, an ophthalmologist, and a radiologist (if indicated). The study protocol was approved by the Institutional Ethics Committee (2022/35-22). A written informed consent form was obtained from the parents and/or the legal guardians of the patients. The study was conducted in accordance with the principles of the Declaration of Helsinki.

Patients referred to our center with vascular malformations that were not IHs were not included. Treatment indication for propranolol was established by our multidisciplinary team. Propranolol was the first choice for treatment, and it was used in tablet form between 2010 and 2021, but as of 2021, we initiated using a suspension form. Before the start of treatment, a careful medical history was recorded, and a detailed physical examination was performed. Contraindications and history of risk factors such as cardiac arrhythmia, reactive airway disease, hypotension, and hypoglycemia were assessed. Patients with these risk factors were excluded. Pretreatment laboratory investigations included a full blood count, glucose, electrolyte, and renal and liver function tests. All patients underwent pediatric cardiology examination with baseline electrocardiography (ECG) and echocardiography if indicated. We followed the patients for side effects such as hypotension, bradycardia, and hypoglycemia during hospitalization. Heart rate and blood pressure were measured before administering the drug, and after two hours, blood glucose levels were measured from the fingertip. We administered the drug orally divided in two doses, starting at a low dose of 0.5 mg/kg/day and gradually increasing to 2 mg/kg/day, and all the patients were hospitalized for 48 to 72 h. The first check-up was performed 7–10 days after discharge from the hospital. If there was no problem, monthly clinical evaluations were performed. At each clinic visit, drug dosage adjustments were made according to the child’s weight. The assessment included physical examination and photographic documentation at each visit. Serial photographs of the patients were taken during the treatment. ECG and laboratory tests were conducted every 3 months. Patients were also evaluated for propranolol side effects through medical histories and physical examinations. Patients with periorbital hemangiomas were also monitored by ophthalmologists, while patients with subglottic IHs were monitored by otolaryngologists. Treatment was discontinued after the proliferation phase. The medication was reduced to four weeks of one daily dose (1 mg/kg). In case of rebound growth, the dose was increased again to twice a day. Treatment was discontinued at the age of 12 to18 months, depending on the localization and ⁄or primary treatment indication. Follow-up visits were performed at one and six months after the cessation of therapy.

Infantile hemangiomas were localized in the head and neck region in 60.5% (248/410) of these cases. The age at the time of diagnosis, sex, history of prematurity, hemangioma type, location, number, complications, indications for treatment, treatment methods (medical and/or surgical), treatment responses, and follow-up times were recorded.

## 3. Statistical Analysis

Statistical analysis was performed using SPSS version 22.0 (IBM Corp., Armonk, NY, USA). Descriptive data are expressed as median (min–max) or number and frequency, where applicable.

## 4. Results

A total of 248 patients with IHs in the head and neck region were followed up in our center. The median age at the time of admission was 4 months (0–13 years), and the female/male ratio (170/78) was 2.18. Among these cases, 64.1% were under six months of age. The age distribution of the cases is shown in Table 1. A prematurity history was present in 28% of patients.

In 72% of the cases (179/248), there were only head and neck IHs, while the remaining 28% had additional trunk and/or extremity IHs. The characteristics of the hemangiomas on admission are shown in Table 2. Figure 1, Figure 2 and Figure 3 show the IH patients.

The follow-up strategy was only applied to 49% (n = 121) of all patients, and 23% (28/121) of these patients did not attend physical examination after the first admission. Those 28 patients were considered to have no complications and were included in the study. Although no treatment was provided, local complications were not observed in any of the patients who regularly attended outpatient clinic examinations. The median follow-up period of these patients was 6 months (0–84 months). None of the patients with spontaneously regressed lesions underwent surgery.

There was an indication for treatment in 55% (136/248) of the cases. Treatment was initiated in 127 (51%) of these, excluding 9 patients. The families of these nine patients did not accept treatment and were followed without treatment. Treatment indications for propranolol were established by a multidisciplinary team. Periorbital localization (visual impairment risk), respiratory distress (subglottic localization, nasal obstruction), perioral localization (nutritional problems), periauricular localization (hearing impairment risk), bleeding, infection, ulceration, and cosmetic risk were the inclusion criteria. Treatment was indicated in complicated IHs. Propranolol was started in all 127 patients with treatment indication. Sixteen patients experienced complications during propranolol therapy (12.5%): fourteen patients had elevated liver function tests, which completely regressed after reducing the dose of the drug. Two patients experienced hypoglycemia during inpatient initiation of therapy; one of these cases resolved following the correct use of the drug after feeding. Another patient had to stop the therapy due to persistent hypoglycemia. Complications occurred at the beginning of therapy. None of the patients presented with bradycardia (<2 SD of normal) or hypotension (<2 SD of normal). Furthermore, there were no cases of sleep disorders, somnolence, diarrhea, agitation, cold extremities, or wheezing. Elevated liver function tests were later performed after starting the therapy. Hypoglycemia occurred at the beginning of therapy.

Propranolol was exclusively administered to 119 patients. There was no adequate response to propranolol in eight patients despite its use for a median of four weeks. In those patients, steroids and other drugs were added (Figure 1). One patient had hemangioma located on one half of the face and extending to the pharyngeal region, and she was treated with steroids, propranolol, and interferon. In a patient with subglottic hemangioma and severe respiratory distress, a response was obtained by administering vincristine and cyclophosphamide in the acute phase, in addition to steroids and propranolol. The median duration of propranolol treatment was 6 months (3–24 months), and treatment responses were achieved in 98% of the cases. We defined a response as a ≥50% improvement within the first month and a ≥90% improvement before discontinuing the drug. The improvement was characterized by visual assessments, as judged by parents and clinicians. Also, an assessment was carried out by a detailed physical examination, documentation of serial photographs, and measurement of superficial skin hemangiomas. Radiological evaluation was made for deep and mixed hemangiomas, if indicated. Ophthalmological examinations for periorbital hemangiomas and otolaryngological examinations for subglottic hemangiomas are important for objective response evaluation.

Three of the patients with periorbital hemangiomas (3/55) had ocular complications. These patients had poor vision and needed glasses. In one patient, reduction surgery was required because of poor vision due to a hemangioma located in the lower eyelid, and steroids were used in addition to propranolol for one month. Afterward, a partial response was achieved, and fibrous scar development was observed when the child was three years old. Subglottic hemangioma cases (n = 9) were followed by the Department of Pediatric Oncology and Otorhinolaryngology in our center; and all received propranolol. A tracheostomy was performed on one patient because of respiratory distress at the time of diagnosis. All patients with subglottic hemangiomas had a complete response to medical treatment.

Complications were observed in 42 (17%) of the 248 patients, and 62% of these patients were between three- and nine-months-old. All but one of them received treatment. Of the cases with complications, 7 had superficial, 7 had deep, and 28 had mixed-type hemangiomas. Hemangioma treatment indications and complications are listed in Table 3. Treatment indications were as follows: periorbital localization (40%), cosmetic problem (24%), perioral localization (14%), respiratory distress (9%), ulceration (6%), periauricular localization (4%), infection (1.5%), and bleeding (1.5%). Complications were among the indications for IH propranolol treatment, and the most common complication in cases receiving a propranolol treatment was ulceration. Respiratory distress, feeding difficulty, bleeding, vision problems, infection, and hearing problems were the other common complications. The median follow-up period for all patients was 14 months (0–118 months).

## 5. Discussion

In the present study, head and neck IHs constituted 60.5% of all hemangiomas. While IHs are seen at a rate of 3 to 10% in infants, their incidence increases up to 22% in those with a history of premature birth [10]. In our study, the rate of prematurity was 28%. Infantile hemangiomas are seen three to five times more frequently in girls than in boys [1,2]. The female/male ratio of our patients was 2.18, which is consistent with the literature.

Hemangiomas can be superficial, deep, or mixed according to their location. About 70 to 90% of them are single hemangiomas [1,2]. Similarly, in 72% of our cases, there were only head and neck IHs, while the remaining 28% had additional trunk and/or extremity IHs.

Although most IHs are expected to regress spontaneously, the treatment approach varies according to the patient. The time of treatment onset is also important to achieve more favorable treatment responses, as better responses can be achieved when initiated in the proliferative phase. In a previous study evaluating all IHs admitted to our center between 2000 and 2007, the complication rate was found to be higher [11]. During that period, only IHs with a high risk of complications were referred to oncology clinics. Complications developed in 35% of patients with head and neck IHs. With the introduction of propranolol as the first-line therapy after 2008, initiating treatment has become more frequent, and the complication rate may have decreased [12,13,14,15,16,17]. Before the serendipitous discovery of propranolol [13], steroids had an important place in the treatment of IHs, particularly in difficult and complicated cases. Their side effect profiles are broad, and their effect is limited. However, since propranolol came into use for airway hemangiomas, complications and the need for surgical interventions such as tracheostomy have almost disappeared [3,17,18]. In our study, nine patients had subglottic hemangiomas, and a response was achieved with propranolol in all these patients. Respiratory distress developed in only one patient in the first week of treatment, and a tracheostomy was performed at the time of diagnosis. However, it resolved after the effective propranolol dose was increased. We also had two patients at our clinic who were followed for subglottic hemangioma before 2008. They required tracheostomies. Propranolol is a synthetic beta-adrenergic receptor blocker. Although the mechanism of action of propranolol has not yet been fully elucidated, it is believed to cause vasoconstriction, decreased renin production, angiogenesis inhibition, and apoptosis stimulation [19,20]. A report related to using propranolol and its effective optimal dose was published in 2013 [12]. The dose can be started as 0.5 mg/kg/day and increased up to 2–3 mg/kg/day, divided into two or three doses. Treatment can be continued for up to 12 to 18 months [12,19,20,21,22]. In our center, treatment begins at 0.5 mg/kg/day and gradually increases to 2 mg/kg/day in two divided doses.

In our study, all patients underwent pediatric cardiology examination with baseline ECG. Echocardiography was performed if indicated. Before starting propranolol, baseline ECG and echocardiogram were not needed in otherwise healthy children in several studies [21]. In a report of consensus conference, the frequency of ECG was noted at 81%, and the frequency of echocardiogram was noted at 38% before starting propranolol [12].

Problematic and complicated cases that require treatment are referred to our experienced tertiary center. Treatment indications for propranolol were established by a multidisciplinary team. Periorbital localization (visual impairment risk), respiratory distress (subglottic localization, nasal obstruction), perioral localization (nutritional problems), periauricular localization (hearing impairment risk), bleeding, infection, ulceration, and cosmetic risk were the inclusion criteria. Treatment was indicated for complicated IHs.

Major local complications are ulceration, bleeding, and infection. Particularly in head and neck IHs, the risk of developing local complications such as bleeding and ulcers determines the need for treatment. Periorbital, perioral, intraoral, and subglottic locations and the compression and obstruction of important organs such as the eyes, mouth, and nose are indications for treatment. Periorbital and eyelid hemangiomas can cause amblyopia, anisometropia, astigmatism, and even blindness [22,23,24,25,26]. Conducting follow-ups in collaboration with an ophthalmologist is of utmost importance from the beginning of the diagnosis. Feeding difficulties can also be seen in perioral hemangiomas. While the complication rate of our patients was 17%, local complications accounted for half of them, consistent with the literature. This was followed by respiratory distress, feeding difficulty, and vision and hearing problems. The ages of 62% of these children were between three to nine months. Léauté-Labrèze et al. [27] recently reported that therapy initiation before 2.5 months of age is associated with a significantly higher rate of treatment success with propranolol. In another recent study, the early (before three months of age) initiation of IH treatment with propranolol resulted in significantly higher response rates [28].

In this study, the most commonly adverse effects of propranolol were found to be hypoglycemia and impaired liver function tests. Hypotension, bradycardia, sleep disturbances, cold extremities, diarrhea, and gastroesophageal reflux are other reported side effects reported in the literature. In order to observe these side effects, patients were hospitalized and monitored at the time of initiation of treatment. Blood glucose levels, blood pressure, and pulse are monitored every 1 to 3 h after taking the drug. The drug should be administered after feeding. Cardiogenic shock, sinus bradycardia, hypotension, heart blocks above the first degree, heart failure, and a history of bronchial asthma are contraindications for propranolol treatment [22,26]. In the present study, hypoglycemia was detected in two cases due to propranolol treatment in the acute period; one of the patients continued on the drug with corrected use after feeding, and the other one had to stop the drug because of persistence. In the follow-up during propranolol use, we observed abnormalities in liver tests (11%) most frequently, which were controlled by dose reductions. None of the patients presented bradycardia (<2 SD of normal) or hypotension (<2 SD of normal). Furthermore, there were no cases of sleep disorders, somnolence, diarrhea, agitation, cold extremities, or wheezing. In some cases, propranolol treatment was interrupted during bronchiolitis periods.

In the first retrospective study related to propranolol, the response rate was 97% [17]. In our study, the response rate to treatment was 98%, and in another study, the response rate to treatment was 97.63% [23]. In another single-center study, all hemangiomas treated with 2 mg/kg/day of propranolol were evaluated, and the treatment response was 96% [15]. In a multicenter, double-blind, randomized study with 495 cases, at the optimal dose of propranolol, the response rate to treatment with 2 to 3 mg/kg/day was >90% [14]. In this study, 1 mg/kg/day was found to be less effective. If there was no adequate response to propranolol despite regular use, oral prednisolone was added for one month and then tapered off, which led to a good response, and the propranolol treatment was resumed [3,25]. In our study, an adequate response was not obtained with propranolol in a patient with a periorbital hemangioma, and an effective response was only achieved by using it in combination with steroids for one month. Histopathological confirmation may be required for hemangiomas when no response is achieved despite propranolol treatment. In one of our cases, with an IH located on the root of the nose, there was no response to propranolol, and the patient was diagnosed with rhabdomyosarcoma after a biopsy [29,30].

Infantile hemangiomas are expected to regress spontaneously. Particularly in superficial IHs, the wait-and-see approach is our center’s preference when there are no complications. Topical timolol is an alternative treatment option for superficial hemangiomas. Topical timolol cases have been reported in the literature. The best response occurred in <1 mm thick hemangiomas [31]. 

After the wait-and-see approach, pharmacotherapy is the mainstay IH treatment. Propranolol has been the first pharmacotherapy option for approximately 15 years. Intralesional steroids, vincristine, and interferon-α have been studied for IH treatment. We have had experience with these options since before 2010. Oral propranolol is more effective and safer than intralesional corticosteroids. Surgical and laser treatments (if necessary), are commonly recommended after 4 to 5 years of age. Laser therapy is a treatment option used for cosmetic purposes, particularly for capillary vascular malformations and in follow-ups of IHs after the regression phase [22].

In our study, complicated and high-risk IHs in the proliferative phase were evaluated. Additionally, it has been reported in the literature that applying laser treatments in the proliferative phase creates a risk of ulceration in the lesion. Although the case reports of laser photocoagulation applied to laryngeal hemangiomas are present in the literature, complication rates for this process in this region have been reported to be high [22].

## 6. Conclusions

In conclusion, IHs are the most common benign tumors of infancy, and more than half of IHs occur in the head and neck region. Although most IHs are expected to regress spontaneously, the main determinants in treatment decisions are the location of the IH and vital functions that may be at risk. The wait-and-see approach is the mainstay for uncomplicated IHs. The timing of treatment is critical. Complications may occur, particularly during the proliferation phase. A multidisciplinary approach is important, particularly in the follow-up of subglottic and periorbital hemangiomas. If IH compromises the vital functions and/or if it is complicated, propranolol is the first-line treatment option due to its safety and effectiveness. Early treatment initiation increases the success rate.

## Figures and Tables

**Figure 1 children-11-00311-f001:**
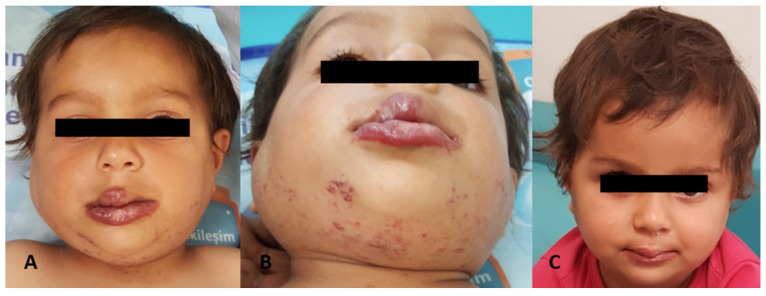
A nine-month-old girl. Parotid, maxillofacial hemangioma treated with pulse methylprednisolone and propranolol (did not respond adequately to propranolol only). (**A**,**B**) before treatment; (**C**) 3 months later.

**Figure 2 children-11-00311-f002:**
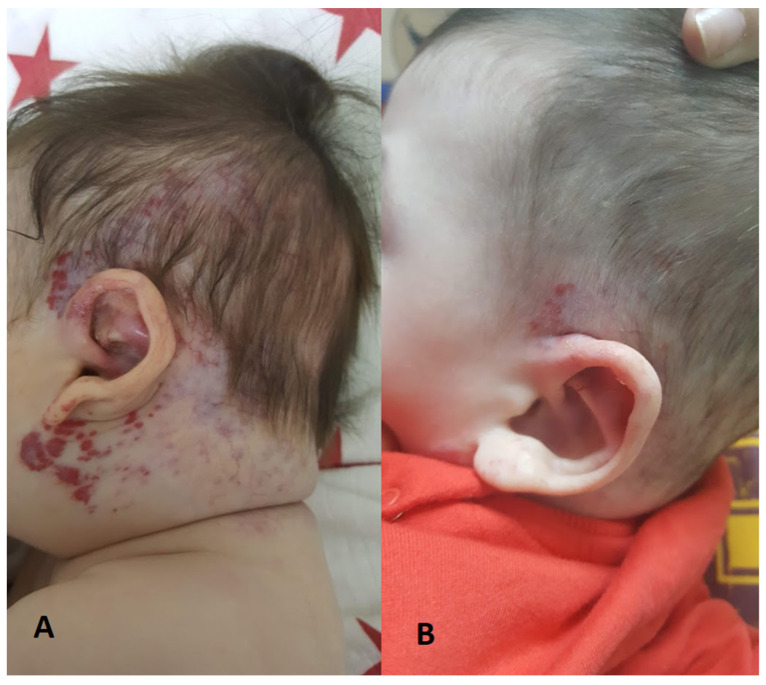
Treatment response to oral propranolol in IH patient. (**A**) before propranolol treatment; (**B**) 3 months later.

**Figure 3 children-11-00311-f003:**
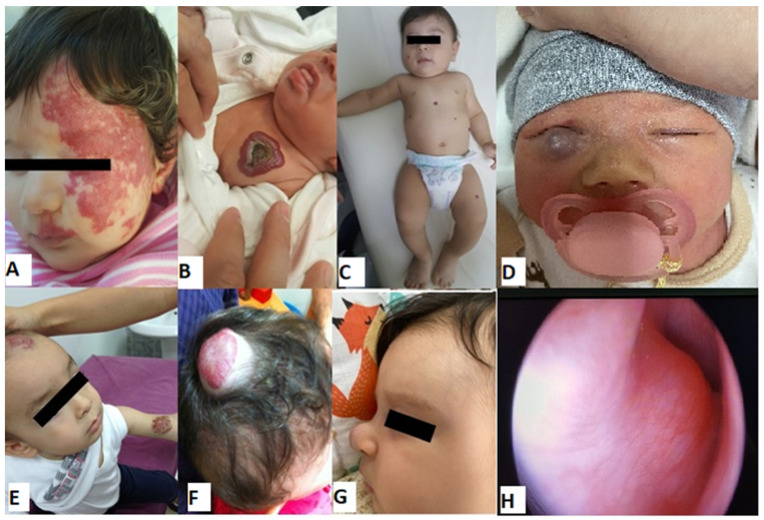
Examples of infantile hemangiomas with indications for treatment (baseline). (**A**) A female patient with PHACES syndrome at 2 months of age. (**B**) A six-week-old female patient with an ulcerated hemangioma of the chest. (**C**) A five-month-old male patient with multifocal hemangioma. (**D**) A six-week-old girl with hemangioma located on the lower eyelid. (**E**) Multifocal mixed IH. (**F**) A one-year-old male patient with a large mixed hemangioma of the scalp and extremities. (**G**) Profile of patient with IH of the nasal tip (cyrano node) at 2 months of age. (**H**) A four-month-old girl with a subglottic hemangioma presenting with stridor.

**Table 1 children-11-00311-t001:** Age distribution of cases.

Age at Admission (Months)	N (%)	Complication N (%)
<3	60 (25)	10 (24)
3–9	133 (54)	26 (62)
>9	55 (22)	6 (14)
**Total**	**248 (100)**	**42 (100)**

**Table 2 children-11-00311-t002:** Characteristics of hemangiomas at diagnosis.

	n	%
Number		
1	150	59
2	53	22
≥2	45	19
Type		
Superficial	118	48
Deep	26	10
Mixed	104	42
Localization		
Scalp	27	11
Forehead	18	7
Periorbital	68	22
Perinasal	24	15
Periauricular	23	9
Cheek	23	9
Perioral/oral	25	10
Parotis	8	3
Subglottic	9	4
Chin	4	2
Neck	13	5
Other face localization	6	3

**Table 3 children-11-00311-t003:** Hemangioma treatment indications and complications.

	n	%
Treatment indications	136	
Periorbital localization	55	40
Respiratory distress	12	9
Perioral localization	19	14
Periauricular localization	6	4
Infection	2	1.5
Bleeding	2	1.5
Ulceration	8	6
Cosmetic	32	24
Complications	42	17
Vision problems	3	7
Hearing problems	1	2
Feeding difficulty	5	12
Respiratory distress	11	26
Infection	2	5
Bleeding	4	10
Ulceration	16	38

## Data Availability

The data that support the findings of this study are available upon request from the corresponding authors. The data are not publicly available because of privacy or ethical restrictions.

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
