# Peer review of "Infantile Hemangiomas of the Head and Neck: A Single-Center Experience"

_children, 2024, doi:10.3390/children11030311_

Round 1

Reviewer 1 Report

Comments and Suggestions for Authors

This is not a novel study.

Please describe all the tests performed before starting propranolol therapy: laboratory tests, ECG, echocardiography perhaps? Did you monitor ECG during scheduled control visits?

Elaborate more about the inclusion criteria for propranolol treatment of IH. Were there any exclusion criteria?

Responses to the propranolol treatment are usually graded as: excellent, good, poor, or no response. Could you distinguish such groups among your patients?

Clearly elucidate the complications of propranolol treatment, and the percentage of patients in which propranolol therapy was discontinued due to complications. Do those complications occur at the beginning of therapy, or some time after starting the therapy?

There are other types of treatment of IH – topical treatment with timolol and also laser therapy. Authors should include this information in discussion section.

The paper needs also some attention to grammar and misspelling e.g.

Abstract: “success rates” change to “success rate”

Background: remove the sentence  “They are often not present at birth and pale pink and indistinctly small lesions at the beginning, which then enter the rapid growth phase and become evident within weeks.”  The same information can be found in the next sentences.

Results: “Only propranolol was administered in 119 of the patients” change to “Exclusive propranolol therapy was administered in…”

“and fibrous scar residue development was observed only after three years of age.” Change to  “and fibrous scar development was observed when the child was three years old”.

“The most common complication in cases receiving propranolol treatment…” Change to “ Among indications for propranolol treatment of IH were complications, the most common…”

Discussion: “Although the mechanism of action in the treatment of his has not been fully elucidated yet..” change to “Although the mechanism of action of propranolol has not been fully elucidated yet…”

This sentence is incomprehensible: “In patients taking propranolol, it may be necessary to add a short-term steroid to control growth at first” – what type of steroid therapy patients needed? I believe that steroid therapy was implemented to control growth of IH?

Conclusions: change “and the vital functions it bears the risk.” to “and life functions that are at risk”

Recommendation: major revision

Comments on the Quality of English Language

The paper needs also some attention to grammar and misspelling e.g.

Author Response

We thank the reviewer for her/his insightful comments. 

As per the recommendations of the rewiever, we have rewritten some parts in more detail, also coreected the sentences, 

Our work has become much more valuable thanks to you.

Response to Reviewer 1 Comments:

We thank the reviewer for taking time to review this manuscript and for his/her insightful comments. As per the recommendations of the reviewer, detailed responses below and the corresponding revisions/corrections highlighted/in track changes in the re-submitted files.

Comments 1: This is not a novel study.
Response 1: Thank you for pointing this.

In our country, infantile hemangiomas are followed by general pediatricians. However, problematic, complicated cases that require treatment are referred to us. We are an experienced tertiary center, standart multidisciplinary professional patient follow-ups and treatments are provided for a large number of patients from all over the country.

Our multidisciplinary team is professional and experienced in its field; depending on the area at risk of hemangiomas located in the head and neck region, professional ophthalmologist, otolaryngologist, audiologist, radiologist and oncologist follow-ups can be done in our center.

Our case series includes subglottic hemangiomas (n:9) that cause airway obstruction and periocular hemangiomas (n:68) where vision is at risk. Case series presenting the treatment results of infantile hemangiomas located in the head and neck region with this diversity and such a large number of cases are limited in the literature. Our work is a study in which the standard approach in terms of treatment and follow-up plan is demonstrated, and the treatment results of a large number of patients from a single center are presented. For these reasons, our work is privileged.

Comments 2: Please describe all the tests performed before starting propranolol therapy: laboratory tests, ECG, echocardiography perhaps? Did you monitor ECG during scheduled control visits?

Response 2: We have modified Material and Methods section to emphasize your comment in more detail.

Pretreatment laboratory investigations included a full blood count, glucose, electrolyte, and renal and liver function tests. All patients underwent pediatric cardiology examination with baseline electrocardiography (ECG) and echocardiography if indicated.

We followed the patients for side effects such as hypotension, bradycardia, and hypoglycemia during hospitalization. Heart rate and blood pressure were measured before administering the drug, and after two hours blood glucose levels were measured from the fingertip.

The first check-up was performed 10 days after discharge from the hospital. If there was no problem, monthly clinical evaluations were performed. At each clinic visit, drug dosage adjustments were made according to the child’s weight. The assessment included physical examination and photographic documentation at each visit; ECG and laboratory tests were conducted every 3 months. Patients were also evaluated for propranolol side effects through medical histories and physical examinations.   

Our patients are infants, we definitely evaluate organ functions before propranolol. We find it appropriate to start the medication under hospitalization by monitoring blood sugar, cardiac ritym and blood pressure.

Similarly, it has been seen in the literature that there are routine protocols for starting the drug and monitoring the patients. This is our center's follow-up and treatment protocol.

Comments 3: Elaborate more about the inclusion criteria for propranolol treatment of IH. Were there any exclusion criteria?

Response 3:

Treatment indications for propranolol were established by a multidisciplinary team. Periorbital localization (visual impairment risk), respiratory distress (subglottic localization, nasal obstruction), perioral localization (nutritional problems), periauricular localization (hearing impairment risk), bleeding, infection, ulceration, and cosmetic risk were the inclusion criteria. Treatment was potentially indicated in complicated infantile hemangiomas.

Before the start of treatment, a careful medical history was recorded, and a detailed physical examination was performed. Contraindications and a history of risk factors such as cardiac arrhythmia, reactive airway disease, hypotension, and hypoglycemia were assessed. Patients with these risk factors were excluded.

This is added to Material and Methods section

Comments 4: Responses to the propranolol treatment are usually graded as: excellent, good, poor, or no response. Could you distinguish such groups among your patients?

Response 4:

We defined a response as a ≥50% improvement within the first month and a ≥90% improvement before discontinuing the drug.

Ophthalmological examinations for periorbital hemangiomas and otolaryngological examinations for subglottic hemangiomas are important for response evaluation.

Comments 5: Clearly elucidate the complications of propranolol treatment, and the percentage of patients in which propranolol therapy was discontinued due to complications. Do those complications occur at the beginning of therapy, or some time after starting the therapy?

Response 5:

Sixteen patients experienced complications during propranolol therapy (12.5%): fourteen patients had elevated liver function tests, which completely regressed after reducing the dose of the drug. Two patients experienced hypoglycemia during inpatient initiation of therapy; one of these cases was resolved following the correct use of the drug after feeding. Another patient had to stop the therapy because of persistent hypoglycemia. Complications occurred at the beginning of therapy. None of the patients presented with bradycardia (< 2 SD of normal) or hypotension (< 2 SD of normal). Furthermore, there were no cases of sleep disorders, somnolence, diarrhea, agitation, cold extremities, or wheezing.

Elevated liver function tests were later performed after starting the therapy. Hypoglycemia occurred at the beginning of therapy.

This information added to the text (Results section).

Comments 6: There are other types of treatment of IH – topical treatment with timolol and also laser therapy. Authors should include this information in discussion section.

Response 6:

Infantile hemangiomas are expected to regress spontaneously. In superficial infantile hemangiomas especially, the wait-and-see approach is our center's preference when there are no complications.

Topical timolol is an alternative treatment option for superficial hemangiomas. Topical timolol cases have been reported in the literature. The best response occurred in <1 mm thick hemangiomas (31). We prefer the wait-and-see approach for uncomplicated superficial head and neck hemangiomas at our center.

After the wait-and-see approach, pharmacotherapy is the mainstay IH treatment. Propranolol has been the first pharmacotherapy option for approximately 15 years.

Intralesional steroids, vincristine, and interferon-α have been studied for IH treatment. We have had experience with these options since before 2010. Oral propranolol is more effective and safer than intralesional corticosteroids. Because of adverse anesthesia effects and blood loss risk, surgical and laser treatments are commonly recommended after 4-5 years of age. Laser therapy is a treatment option used for cosmetic purposes, especially for capillary vascular malformations and in follow-ups of infantile hemangiomas after the regression phase (22).

In our study, cases where infantile hemangiomas were in the proliferative phase and complicated or compromised vital functions were evaluated. Additionally, it has been reported in the literature that applying laser treatments in the proliferative phase creates a risk of ulceration in the lesion. Although case reports of laser photocoagulation applied to laryngeal hemangiomas are present in the literature, complication rates for this process in this region have been reported to be high (22).

Comments 7: The paper needs also some attention to grammar and misspelling e.g.

Response 7: A rapid English editing was done   (English editing ID: english-77404)

Other comments:

  • Abstract: “success rates” change to “success rate”

“success rates” change to “success rate”

  • Background: remove the sentence  “They are often not present at birth and pale pink and indistinctly small lesions at the beginning, which then enter the rapid growth phase and become evident within weeks.”  The same information can be found in the next sentences.

The sentence  “They are often not present at birth and pale pink and indistinctly small lesions at the beginning, which then enter the rapid growth phase and become evident within weeks.”  removed from the text

  • Results: “Only propranolol was administered in 119 of the patients” change to “Exclusive propranolol therapy was administered in…”

The part of sentence “Only propranolol was administered in 119 of the patients” change to “Propranolol was exclusively administered to 119 patients.” in the results section of the text.

  • “and fibrous scar residue development was observed only after three years of age.” Change to  “and fibrous scar development was observed when the child was three years old”.

The part of sentence “and fibrous scar residue development was observed only after three years of age.” change to  “and fibrous scar development was observed when the child was three years old”. in the text.

  • “The most common complication in cases receiving propranolol treatment…” Change to “ Among indications for propranolol treatment of IH were complications, the most common…”

The part of sentence “The most common complication in cases receiving propranolol treatment…” change to “ Complications were among the indications for IH propranolol treatment, and the most common complication in cases receiving a propranolol treatment was ..” in the text.

  • Discussion: “Although the mechanism of action in the treatment of his has not been fully elucidated yet..” change to “Although the mechanism of action of propranolol has not been fully elucidated yet…”

The part of sentence “Although the mechanism of action in the treatment of his has not been fully elucidated yet..” change to “Although the mechanism of action of propranolol has not yet been fully elucidated,…” in the discussion section of the text.

  • This sentence is incomprehensible: “In patients taking propranolol, it may be necessary to add a short-term steroid to control growth at first” – what type of steroid therapy patients needed? I believe that steroid therapy was implemented to control growth of IH?

The incomprehensible sentence changed as: “If there was no adequate response to propranolol despite regular usage, oral prednisolone was added for one month and then tapered off, which led to a good response, and the propeanolol treatment was resumed. ”

  • Conclusions: change “and the vital functions it bears the risk.” to “and life functions that are at risk”

The part of sentence, “and the vital functions it bears the risk.” change to “and life functions that may be at risk” in the conclusions section of the text.

Reviewer 2 Report

Comments and Suggestions for Authors

This manuscript describes the authors’ experience with 248 patients with infantile hemangiomas (IH) of the head and neck, 127 (51%) of whom were treated with propranolol.

I have several observations, questions and suggestions for the authors, which may improve this paper.

1.     The manuscript needs significant revision by an expert in English.  There are multiple errors involving grammar, syntax, and sentence structure.

2.     My biggest concern is that this paper does not present anything new or noteworthy about infantile hemangiomas of the head and neck. The conclusions on the final page are conclusions from other studies, but not this one.

3.     The abstract and discussion say the treatment response rate was 98%.  However, the results report a 93% response rate. 

4.     How was response was defined? More information needs to be provided. 

5.     The ISSVA calcifications are not modified annually. 

6.     The first line under materials and methods needs revision (“XXX”).

7.     Figure 1 should be deleted.  This manuscript does not deal with other vascular malformations besides IH, and this figure does not enhance the manuscript.

8.     For patient who are hospitalized, did you follow a protocol for monitoring?  For example, how often were blood pressure and glucose checked? Can data be provided? 

9.     “After leaving the hospital, we performed monthly check-up calls within 10-day intervals in the first month.” This is confusing and needs to be rewritten. In addition, it is hard to believe that this actually happened.  That would take a tremendous amount of resources and also rely heavily on parents actually answering their phones. Can the authors provide data on how many families were successfully reached by telephone?

1    In several places the word “admission” is used when “presentation” or “diagnosis” is probably a better choice. 

1   Table 1 is not helpful, because the age at “admission” depends on the age of referral.

1     In Table 2 the types of hemangiomas are listed as “capillary, deep, and mix” (which should be “mixed”). “Capillary” should be changed to “superficial,” because all hemangiomas are capillary.  Also, the final line of Table 2 should say “other face location.”

  How many patients had PHACE syndrome (suspected, possible, or definite)?

1    In the results, it says that “Propranolol was started in all patients (n = 127) with indications for treatment.” This is hard to believe.  Indications for treatment are often subjective.  In addition, in my experience, parents often decline treatment when it is offered or recommended.  Conversely, sometimes parents request treatment even if it may not be indicated or recommended. Shared decision making should be employed.

  Can the authors provide data on how many patients (parents) discontinued propranolol?

  Was topical timolol used or offered to any patients?

  More information should be provided about the abnormal liver function tests and the two cases of hypoglycemia which "resolved following the correct use of the drug after feeding."

  The fact that sleep disturbances and diarrhea were not observed indicates a bad study design and that side effects were not systematically and routinely assessed. Almost all children have sleep disorders or diarrhea at some point (as well as other symptoms, whether they are taking propranolol or not).

  Into the Results, “infection/bleeding/ulceration” are lumped together in the text and also Table 3.  Why are they lumped together?  Those are 3 separate and important complications. 

2    In the Discussion "A report related to the use of propranolol and its effective optimal dose was published in 2013 (22)”.  This is the wrong citation. Please check all citations for accuracy.

  "The most commonly reported adverse effects of propranolol are hypotension, hypoglycemia, asymptomatic bradycardia, impaired liver function test, and hyperkalemia.”  This sentence is wrong in many aspects. Hypotension hypoglycemia, impaired liver function tests, and hyperkalemia are very rare in every large study which systematically tracked side effects.

  Infants less than 8 weeks old are not necessarily required to be hospitalized.  In addition, baseline EKG and echocardiogram are not needed in otherwise healthy children (several studies have shown this to be true).

  “Other than that, no side effects were observed.”  Again, this speaks to a lack of thorough evaluation and side effect tracking.

Comments on the Quality of English Language

See detailed comments above.

Author Response

We thank the reviewer for her/his insightful comments. 

As per the recommendations of the rewiever, we have rewritten some parts in more detail, also coreected the sentences, 

Our work has become much more valuable thanks to you.

Response to Reviewer 2 Comments:

We thank the reviewer for taking time to review this manuscript and for his/her insightful comments. As per the recommendations of the reviewer, detailed responses below and the corresponding revisions/corrections highlighted/in track changes in the re-submitted files.

Comment 1: The manuscript needs significant revision by an expert in English.  There are multiple errors involving grammar, syntax, and sentence structure.

Response 1: A rapid English editing was done   (English editing ID: english-77404)

Comment 2: My biggest concern is that this paper does not present anything new or noteworthy about infantile hemangiomas of the head and neck. The conclusions on the final page are conclusions from other studies, but not this one.

Response 2: In our country, infantile hemangiomas are followed by general pediatricians. However, problematic, complicated cases that require treatment are referred to us. We are an experienced tertiary center, standart multidisciplinary professional patient follow-ups and treatments are provided for a large number of patients from all over the country.

Our multidisciplinary team is professional and experienced in its field; depending on the area at risk of hemangiomas located in the head and neck region, professional ophthalmologist, otolaryngologist, audiologist, radiologist and oncologist follow-ups can be done in our center.

Our case series includes subglottic hemangiomas (n:9) that cause airway obstruction and periocular hemangiomas (n:68) where vision is at risk. Case series presenting the treatment results of infantile hemangiomas located in the head and neck region with this diversity and such a large number of cases are limited in the literature. Our work is a study in which the standard approach in terms of treatment and follow-up plan is demonstrated, and the treatment results of a large number of patients from a single center are presented. For these reasons, our work is privileged.

      Comment 3: The abstract and discussion say the treatment response rate was 98%.  However, the results report a 93% response rate. 

      Response 3: The response rate corrected as 98% in the results section.

      Comment 4: How was response was defined? More information needs to be provided. 

Response 4: We defined a response as a ≥50% improvement within the first month and a ≥90% improvement before discontinuing the drug.

Ophthalmological examinations for periorbital hemangiomas and otolaryngological examinations for subglottic hemangiomas are important for response evaluation.

Comment 5: The ISSVA clssiffications are not modified annually. 

      Response 5: The ISSVA classifications are not modified annually, but are improved at regular intervals. (https://www.issva.org/classification).

      Comment 6: The first line under materials and methods needs revision (“XXX”).

      Response 6: The first line under materials and methods revised: XXX to Dokuz Eylül University

      Comment 7: Figure 1 should be deleted.  This manuscript does not deal with other vascular malformations besides IH, and this figure does not enhance the manuscript.

      Response 7: Figure 1 deleted.

      Comment 8: For patient who are hospitalized, did you follow a protocol for monitoring?  For example, how often were blood pressure and glucose checked? Can data be provided? 

Response 8: Material and methods section was written in more detail.

Before the start of treatment, a careful medical history was recorded, and a detailed physical examination was performed. Contraindications and a history of risk factors such as cardiac arrhythmia, reactive airway disease, hypotension, and hypoglycemia were assessed. Patients with these risk factors were excluded. Pretreatment laboratory investigations included a full blood count, glucose, electrolyte, and renal and liver function tests. All patients underwent pediatric cardiology examination with baseline electrocardiography (ECG) and echocardiography if indicated.

We followed the patients for side effects such as hypotension, bradycardia, and hypoglycemia during hospitalization. Heart rate and blood pressure were measured before administering the drug, and after two hours blood glucose levels were measured from the fingertip.

Our patients are infants, we definitely evaluate organ functions before propranolol. We find it appropriate to start the medication under hospitalization by monitoring blood sugar, cardiac ritym and blood pressure.

Similarly, it has been seen in the literature that there are routine protocols for starting the drug and monitoring the patients. This is our center's follow-up and treatment protocol.

      Comment 9: “After leaving the hospital, we performed monthly check-up calls within 10-day intervals in the first month.” This is confusing and needs to be rewritten. In addition, it is hard to believe that this actually happened.  That would take a tremendous amount of resources and also rely heavily on parents actually answering their phones. Can the authors provide data on how many families were successfully reached by telephone?

Response 9: The first check-up was performed 10 days after discharge from the hospital. If there was no problem, monthly clinical evaluations were performed. At each clinic visit, drug dosage adjustments were made according to the child’s weight. The assessment included physical examination and photographic documentation at each visit; ECG and laboratory tests were conducted every 3 months. Patients were also evaluated for propranolol side effects through medical histories and physical examinations.    

      There was no problem with our patients coming for routine check-ups and there was no need to reach them by phone.

      Comment 10: In several places the word “admission” is used when “presentation” or  

      “diagnosis” is probably a better choice. 

      Response 10: The word ‘admission’ is changed to ‘diagnosis’ in appropriate places.

Comment 11: Table 1 is not helpful, because the age at “admission” depends on the age of referral.

Response 11: The admission age depends on the age of referral. This is nearly same as the age at the beginning of therapy. As we had a large number of patients, we thought it would be appropriate to classify them according to their age at presentation.

Comment 12: In Table 2 the types of hemangiomas are listed as “capillary, deep, and mix” (which should be “mixed”). “Capillary” should be changed to “superficial,” because all hemangiomas are capillary.  Also, the final line of Table 2 should say “other face location.”

Response 12: In the text and Table 2,the term capillary changed to superficial. Also, the final line of Table 2 changed to “other face location.”

Comment 13: How many patients had PHACE syndrome (suspected, possible, or definite)?

Response 13: Only one patient had suspected PHACE syndrome.

Comment 14: In the results, it says that “Propranolol was started in all patients (n = 127) with indications for treatment.” This is hard to believe.  Indications for treatment are often subjective.  In addition, in my experience, parents often decline treatment when it is offered or recommended.  Conversely, sometimes parents request treatment even if it may not be indicated or recommended. Shared decision making should be employed.

Response 14: There was an indication for treatment in 55% (136/248) of the cases. Treatment was initiated in 127 (51%) of these, excluding 9 patients. The families of these nine patients did not accept treatment and were followed without treatment.

As we are an experienced tertiary center, problematic and complicated cases that require treatment are referred to us.

Treatment indications for propranolol were established by a multidisciplinary team, Periorbital localization (visual impairment risk), respiratory distress (subglottic localization, nasal obstruction), perioral localization (nutritional problems), periauricular localization (hearing impairment risk), bleeding, infection, ulceration and cosmetic risk are the inclusion criteria. Treatment was potentially indicated in complicated infantile hemangiomas.

Our experience is added to discussion part.

Comment 15: Can the authors provide data on how many patients (parents) discontinued propranolol?

Response 15: None of the patients discontinued the treatment in our study.

Comment 16: Was topical timolol used or offered to any patients?

Response 16: Infantile hemangiomas are expected to regress spontaneously. In superficial infantile hemangiomas especially, the wait-and-see approach is our center's preference when there are no complications.

Topical timolol is an alternative treatment option for superficial hemangiomas. Topical timolol cases have been reported in the literature. The best response occurred in <1 mm thick hemangiomas (31). We prefer the wait-and-see approach for uncomplicated superficial head and neck hemangiomas at our center.

Comment 17: More information should be provided about the abnormal liver function tests and the two cases of hypoglycemia which "resolved following the correct use of the drug after feeding."

Response 17: Sixteen patients experienced complications during propranolol therapy (12.5%): fourteen patients had elevated liver function tests, which completely regressed after reducing the dose of the drug. Two patients experienced hypoglycemia during inpatient initiation of therapy; one of these cases was resolved following the correct use of the drug after feeding. Another patient had to stop the therapy because of persistent hypoglycemia. Complications occurred at the beginning of therapy. None of the patients presented with bradycardia (< 2 SD of normal) or hypotension (< 2 SD of normal). Furthermore, there were no cases of sleep disorders, somnolence, diarrhea, agitation, cold extremities, or wheezing.

Elevated liver function tests were later performed after starting the therapy. Hypoglycemia occurred at the beginning of therapy.

This information added to the text (Results section).

Comment 18: The fact that sleep disturbances and diarrhea were not observed indicates a bad study design and that side effects were not systematically and routinely assessed. Almost all children have sleep disorders or diarrhea at some point (as well as other symptoms, whether they are taking propranolol or not).

Response 18: Yes, you are right, the drug may have such side effects, and these were systematically and routinely assessed at each clinical visit, the drug dosage adjustments made according to the child’s weight. The assessment included medical history, physical examination and photographic documentation.

Sleep disturbances, diarrhea, respiratory problems, hypoglycemia, agitation assessed by medical history and  physical examination. ECG and laboratuary tests performed every 3 months.

None of the cases presented with sleep disturbances related to propranolol, and no one applied with such a complaint.

We did not detect any propranolol-related diarrhea.

Comment 19: Into the Results, “infection/bleeding/ulceration” are lumped together in the text and also Table 3.  Why are they lumped together?  Those are 3 separate and important complications. 

Response 19: Infection/bleeding/ulceration are seperated for treatment indications and hemangioma complications, in the text and Table.

Comment 20: In the Discussion "A report related to the use of propranolol and its effective optimal dose was published in 2013 (22)”.  This is the wrong citation. Please check all citations for accuracy.

Response 20: All citations checked again, in the Discussion "A report related to the use of propranolol and its effective optimal dose was published in 2013 is not 22, it is 12, it is corrected. 

Comment 21: "The most commonly reported adverse effects of propranolol are hypotension, hypoglycemia, asymptomatic bradycardia, impaired liver function test, and hyperkalemia.”  This sentence is wrong in many aspects. Hypotension hypoglycemia, impaired liver function tests, and hyperkalemia are very rare in every large study which systematically tracked side effects.

Response 21: As we are an experienced tertiary center, problematic cases referred to us. Standard multidisciplinary professional patient follow-ups and treatments are provided for a large number of patients.

The most commonly reported adverse effects of propranolol are hypotension, hypoglycemia, bradycardia, impaired liver function tests, and hyperkalemia. Sleep disturbances, cold extremities, diarrhea, and gastroesophageal reflux are other reported side effects. To prevent these side effects, small babies (<8 weeks old) are hospitalized to start treatment. Blood glucose levels, blood pressure, and pulse are monitored every 1 to 3 hours after taking the drug. The drug should be administered after feeding. Baseline electrocardiography and echography are performed before treatment. Cardiogenic shock, sinus bradycardia, hypotension, heart blocks above the first degree, heart failure, and a history of bronchial asthma are contraindications for propranolol treatment (22, 26). In the present study, hypoglycemia was detected in two cases because of propranolol treatment in the acute period; one of the patients continued on the drug with corrected use after feeding, and the other one had to stop the drug because of persistence. In the follow-up during propranolol use, we observed abnormalities in liver tests (11%) most frequently, which were controlled by dose reductions. None of the patients presented bradycardia (< 2 SD of normal) or hypotension (< 2 SD of normal). Furthermore, there were no cases of sleep disorders, somnolence, diarrhea, agitation, cold extremities, or wheezing. In some cases, propranolol treatment was interrupted during bronchiolitis periods.

Comment 22: Infants less than 8 weeks old are not necessarily required to be hospitalized.  In addition, baseline EKG and echocardiogram are not needed in otherwise healthy children (several studies have shown this to be true).

Response 22: As several studies have shown patients are not necessarily required to be hospitalized,  we are an experienced tertiary center, problematic and complicated cases that require treatment are referred to us. Our patients are infants, we definitely evaluate organ functions before propranolol. We find it appropriate to start the medication under hospitalization by monitoring blood sugar, cardiac ritym and blood pressure.

Similarly, it has been seen in the literature that there are routine protocols for starting the drug and monitoring the patients. This is our center's follow-up and treatment protocol. We find hospitalization appropriate due to the age group of the patients.

In addition to studies arguing that hospitalization is not necessary, case series in which treatment was initiated by hospitalization have also been reported in the literature (Stringari G, Barbato G, Zanzucchi M, Marchesi M, Cerasoli G, Tchana B, et al. Propranolol treatment for infantile hemangioma: a case series of sixty-two patients. Pediatr Med Chir. 2016 Jun 27;38(2):113. doi: 10.4081/pmc.2016.113.)

      Comment 23:  “Other than that, no side effects were observed.”  Again, this speaks to a lack of thorough evaluation and side effect tracking.

Response 23: None of the patients presented with bradycardia (< 2 SD of normal) or hypotension (< 2 SD of normal). Furthermore, there were no cases of sleep disorders, somnolence, diarrhea, agitation, cold extremities, or wheezing. Elevated liver function tests were later performed after starting the therapy. Hypoglycemia occurred at the beginning of therapy.

Round 2

Reviewer 1 Report

Comments and Suggestions for Authors

The paper can be published in the present form

Author Response

First of all, thank you very much for your careful and detailed comments. As per the recommendations of the reviewer, detailed responses below and the corresponding revisions/corrections highlighted/in track changes in the re-submitted files.

Major:

  1. My biggest concern is that this paper does not present anything new or noteworthy about infantile hemangiomas of the head and neck. Can the authors state what is unique, new, or different about this cohort of patients, compared to what has already been published? What “take home” or key message should be conveyed to readers?

Thank you for your comments,

Failure to present any new discoveries or information about infantile hemangiomas, which have been studied extensively, is a justified criticism of our study. However, through to this study, we wanted to draw attention to head and neck hemangiomas for physicians or readers interested in child health and diseases. In our study, which included a considerable number of patients compared to the literature, cases with serious risks such as airway obstruction and vision problems were reported.

Our case series includes subglottic hemangiomas (n:9) that cause airway obstruction and periocular hemangiomas (n:68) where vision is at risk. Case series presenting the treatment results of infantile hemangiomas located in the head and neck region with this diversity and such a large number of cases are limited in the literature. Our work is a study in which the standard approach in terms of treatment and follow-up plan is demonstrated, and the treatment results of a large number of patients from a single center are presented. For these reasons, our work is privileged.

The unique, new, or different about this cohort of patients:

We underlined the wait-and-see approach is the mainstay of treatment.

We emphasized the importance of a multidisciplinary approach, especially in the follow-up of special ones such as subglottic and periorbital hemangiomas.

Take home messages:

As the infantile hemangiomas are the most common benign vascular tumors of infancy, wait-and-see approach is approciate for uncomplicated infantile hemangiomas.

If the infantile hemangioma compromised vital functions and was complicated, propranolol was given as first-line treatment. Oral propranolol is the first option because of its safety and effectiveness. We observed few and controllable side effects and achieved response in a high rate of 98% of the patients.

Infantile hemangiomas located in the head and neck region such as subglottic and periorbital should be treated with a multidisciplinary approach in an experienced center.

  1. Materials and Methods (and also Discussion): “We administered the drug orally in two doses, starting at a low dose of 0.25 mg/kg/dose and gradually increasing to 2 mg/kg/day…” In the Discussion section (second paragraph), “dose” and “day” are both used in a confusing manner. In particular, the last sentence of theat paragraph is incorrect (at least I hope it is incorrect): “At our center, treatment begins at 0.25 mg/kg/dose and gradually increases to 2 mg/kg/dose.”  I hope the authors mean 2 mg/kg/day and not 2 mg/kg/dose. Please rewrite to be consistent and precise with the dosing used. Specifically, please do not alternate between “dose” and “day.” Pick one and stick with it. If “day” is used, please specify how the total daily dose was divided.

Thank you for your comment, we rewrite the sentences about drug doses. We mean 2 mg/kg/day, you are right:

We administered the drug orally divided in two doses, starting at a low dose of 0.5 mg/kg/day and gradually increasing to 2 mg/kg/day, and all the patients were hospitalized for 48 to 72 hours

  1. Results: “…treatment responses were achieved in 98%. We defined a response as a ≥50% improvement within the first month and a ≥90% improvement before discontinuing the drug.” A 98% response rate, using the definitions supplied (a ≥50% improvement within the first month and a ≥90% improvement before discontinuing the drug”) is hard to believe. How were these percentages measured? For the majority of IH responses, the response is simply judged by visual assessment. A “50%” or “90%” improvement is not accurate to report without objective measurements. I recommend rewriting this important section for clarity. Note: It would be acceptable to report responses based on visual assessments, as judged by parents or providers. But sufficient information would need to be provided (not just made up).

Thank you for pointing this, we have rewritten this part in more detail to clarify:

Treatment responses were achieved in 98% of the cases. We defined a response as a ≥50% improvement within the first month and a ≥90% improvement before discontinuing the drug. Improvement was characterized by visual assessments as judged by parents and clinicians. Also, assessment made by detailed physical examination, documentation of serial photographs, measurement of superficial skin hemangiomas. Radiological evaluation made for deep and mixed hemangiomas, if indicated. Ophthalmological examinations for periorbital hemangiomas and otolaryngological examinations for subglottic hemangiomas are important for objective response evaluation.

  1. The Discussion is much too long. Much of the content in the Discussion is repetitive (such as repeating results) and can be deleted. The Discussion should focus on the points pertinent to the results of this study, not other studies. It reads more like a review article than a research article. If the authors wish to make this a review article (which I do not recommend), that should be stated in the title and the manuscript formatted appropriately.

Thank you for your comments. This is a research article. Experiences of a tertiary center with a large number of patients presented in this study. Follow-up with professional multidisciplinary team is highlighted.

The discussion section revised from this perspective.

Minor:

  1. Background, second sentence: What does “(3:1)” mean? Please rewrite or delete.

“(3:1)” deleted from Background.

  1. Background: “Surgery is often performed for reconstruction after the fatty fibrous residue phase.” This is incorrect. Surgery is rarely performed on residual IHs. I suggest changing “often” to “rarely” or “sometimes.”

‘’often’’ changed to ‘’sometimes’’:

 “Surgery is sometimes performed for reconstruction after the fatty fibrous residue phase.”

  1. Background: The ISSVA classification was most recently updated in 2018, not 2014 (Reference: https://www.issva.org/classification).

The year changed to 2018.

  1. Materials and Methods: “January 1st” should be “January 1.”

                “January 1st” changed to “January 1.”

  1. Materials and Methods: “Cases of hemangioma referred to our center for treatment consistent with vascular malformation were not included.” I suggest changing this sentence to something like “Patients referred to our center with vascular malformations that were not IHs were not included.”

“Cases of hemangioma referred to our center for treatment consistent with vascular malformation were not included.” chnged as “Patients referred to our center with vascular malformations that were not IHs were not included.”

  1. Materials and Methods: “The first check-up was performed 10 days after discharge from the hospital.” I am sure 10 days was not universally applied. A more realistic follow up time would be a range such as 7 – 10 days, or 7 – 12 days. Please clarify.

10 days changed to 7-10 days:

“The first check-up was performed 7-10 days after discharge from the hospital.”

  1. Table 1: “9>” should be “>9”

“9>” changed to “>9”

  1. Results: “Two patients experienced hypoglycemia during inpatient initiation of therapy; one of these cases was resolved following the correct use of the drug after feeding.” Please delete “was.”

“was.” Deleted

  1. Results: “In these patients, steroids and other drugs were added (Figure 2).” Figure 2 should be Figure 1.

Figure 2 changed to Figure 1.

  1. Discussion: “While IHs are seen at a rate of 8 to 12% in infants…” The Background says “3 and 10% in infants under 12 months of age.” Please clarify this discrepancy.

“8 to 12%” changed to  “3 and 10%”

  1. Discussion: “The female/male ratio of our patients was 2.23” The Results say 2.17 (which really should be 2.18 (170/78 = 2.18)). Please clarify this discrepancy.

The female/male ratio corrected as 2.18 in Results and Discussion.

  1. Discussion: “Treatment was potentially indicated for complicated infantile hemangiomas.” Please delete “potentially.” Or delete this whole sentence, which is self-evident.

The word “potentially” deleted.

  1. Discussion: “Major uncontrollable local complications are bleeding, ulcers, and infection.” Please delete “uncontrollable.”

The word “uncontrollable” deleted.

  1. Discussion: “The most commonly reported adverse effects of propranolol are hypotension, hypoglycemia, asymptomatic bradycardia, impaired liver function test, and hyperkalemia.” This sentence is wrong in many aspects. Hypotension, hypoglycemia, impaired liver function tests, and hyperkalemia are rare in every large study which systematically tracked side effects. They are not “the most commonly reported adverse effects.” Please note that this exact comment was made in my original review.

The sentence “The most commonly reported adverse effects of propranolol are hypotension, hypoglycemia, asymptomatic bradycardia, impaired liver function test, and hyperkalemia.”  Changed to “In this study, the most commonly adverse effects of propranolol are hypotension, hypoglycemia, asymptomatic bradycardia, impaired liver function test, and hyperkalemia.” The side effects   hypotension, hypoglycemia, and bradycardia are the known side effects (References 12, 14).

  1. Discussion: “Baseline electrocardiography and echography are performed before treatment.” Baseline EKG and echocardiogram are not needed in otherwise healthy children. Several studies have shown this to be true. Please acknowledge this and include at least one citation.

Baseline EKG and echocardiogram are not needed in otherwise healthy children. Several studies have shown this to be true (Reference 21).

But like the majority of studies, all patients underwent pediatric cardiology examination with baseline electrocardiography (ECG) and echocardiography if indicated in our study (References 22,24). Before starting propranolol, frequency of ECG was noted 81%, echocardiogram was noted 38% in report of a consensus conference (Reference 12).

              Discussion revised for this recommandations.

  1. Discussion (line 253): “In a multicenter, double-blind, randomized study with 495 cases, at the optimal dose of propranolol, the response rate to treatment with 2 to 3 mg/kg/dose was >90% (14). In this study, 1 mg/kg/dose was found to be less effective.” “Dose” should be “day” (in two instances). Also, please change “this” to “that.”

“Dose” changed to “day” (in two instances). Also, “this” changed to “that.”

  1. Discussion: “Because of adverse anesthesia effects and blood loss risk, surgical and laser treatments are commonly recommended after 4-5 years of age.” This statement is not correct. Surgical and laser treatments are rarely used for IH after 4-5 years of age (or any age). Please clarify.

This sentence revised as “Surgical and laser treatments (if necessary), are commonly recommended after 4-5 years of age.” 

  1. Discussion: “In our study, cases where infantile hemangiomas were in the proliferative phase and complicated or compromised vital functions were evaluated.” This sentence is confusing and should be rewritten for clarity, or deleted.

This sentence is rewritten.

  1. Discussion: “As we are an experienced tertiary center, standard multidisciplinary professional patient follow-ups and treatments are provided for a large number of patients.” This sentence should be deleted.

This sentence revised in both resuts and discussion sections.

Reviewer 2 Report

Comments and Suggestions for Authors

This manuscript is improved from its original version. However, I have still have several observations, questions and suggestions for the authors, which may improve this paper.

Major:

1.     My biggest concern is that this paper does not present anything new or noteworthy about infantile hemangiomas of the head and neck. Can the authors state what is unique, new, or different about this cohort of patients, compared to what has already been published? What “take home” or key message should be conveyed to readers?

2.     Materials and Methods (and also Discussion): “We administered the drug orally in two doses, starting at a low dose of 0.25 mg/kg/dose and gradually increasing to 2 mg/kg/day…” In the Discussion section (second paragraph), “dose” and “day” are both used in a confusing manner. In particular, the last sentence of theat paragraph is incorrect (at least I hope it is incorrect): “At our center, treatment begins at 0.25 mg/kg/dose and gradually increases to 2 mg/kg/dose.”  I hope the authors mean 2 mg/kg/day and not 2 mg/kg/dose.    

Please rewrite to be consistent and precise with the dosing used. Specifically, please do not alternate between “dose” and “day.” Pick one and stick with it. If “day” is used, please specify how the total daily dose was divided.

3.     Results: “…treatment responses were achieved in 98%. We defined a response as a ≥50% improvement within the first month and a ≥90% improvement before discontinuing the drug.” A 98% response rate, using the definitions supplied (a ≥50% improvement within the first month and a ≥90% improvement before discontinuing the drug”) is hard to believe. How were these percentages measured? For the majority of IH responses, the response is simply judged by visual assessment. A “50%” or “90%” improvement is not accurate to report without objective measurements. I recommend rewriting this important section for clarity. Note: It would be acceptable to report responses based on visual assessments, as judged by parents or providers. But sufficient information would need to be provided (not just made up).

4.     The Discussion is much too long. Much of the content in the Discussion is repetitive (such as repeating results) and can be deleted. The Discussion should focus on the points pertinent to the results of this study, not other studies. It reads more like a review article than a research article. If the authors wish to make this a review article (which I do not recommend), that should be stated in the title and the manuscript formatted appropriately.

Minor:

1.     Background, second sentence: What does “(3:1)” mean? Please rewrite or delete.

2.     Background: “Surgery is often performed for reconstruction after the fatty fibrous residue phase.” This is incorrect. Surgery is rarely performed on residual IHs. I suggest changing “often” to “rarely” or “sometimes.”

3.     Background: The ISSVA classification was most recently updated in 2018, not 2014 (Reference: https://www.issva.org/classification).  

4.     Materials and Methods: “January 1st” should be “January 1.”

5.     Materials and Methods: “Cases of hemangioma referred to our center for treatment consistent with vascular malformation were not included.” I suggest changing this sentence to something like “Patients referred to our center with vascular malformations that were not IHs were not included.”

6.     Materials and Methods: “The first check-up was performed 10 days after discharge from the hospital.” I am sure 10 days was not universally applied. A more realistic follow up time would be a range such as 7 – 10 days, or 7 – 12 days. Please clarify.

7.     Table 1: “9>” should be “>9”

8.     Results: “Two patients experienced hypoglycemia during inpatient initiation of therapy; one of these cases was resolved following the correct use of the drug after feeding.” Please delete “was.”

9.     Results: “In these patients, steroids and other drugs were added (Figure 2).” Figure 2 should be Figure 1.

10.  Discussion: “While IHs are seen at a rate of 8 to 12% in infants…” The Background says “3 and 10% in infants under 12 months of age.” Please clarify this discrepancy.

11.  Discussion: “The female/male ratio of our patients was 2.23” The Results say 2.17 (which really should be 2.18 (170/78 = 2.18)). Please clarify this discrepancy.

12.  Discussion: “Treatment was potentially indicated for complicated infantile hemangiomas.” Please delete “potentially.” Or delete this whole sentence, which is self-evident.

13.  Discussion: “Major uncontrollable local complications are bleeding, ulcers, and infection.” Please delete “uncontrollable.”

14.  Discussion: “"The most commonly reported adverse effects of propranolol are hypotension, hypoglycemia, asymptomatic bradycardia, impaired liver function test, and hyperkalemia.”  This sentence is wrong in many aspects. Hypotension, hypoglycemia, impaired liver function tests, and hyperkalemia are rare in every large study which systematically tracked side effects. They are not “the most commonly reported adverse effects.” Please note that this exact comment was made in my original review.

15.  Discussion: “Baseline electrocardiography and echography are performed before treatment.”  Baseline EKG and echocardiogram are not needed in otherwise healthy children. Several studies have shown this to be true. Please acknowledge this and include at least one citation.

16.  Discussion (line 253): “In a multicenter, double-blind, randomized study with 495 cases, at the optimal dose of propranolol, the response rate to treatment with 2 to 3 mg/kg/dose was >90% (14). In this study, 1 mg/kg/dose was found to be less effective.” “Dose” should be “day” (in two instances). Also, please change “this” to “that.”

17.  Discussion: “Because of adverse anesthesia effects and blood loss risk, surgical and laser treatments are commonly recommended after 4-5 years of age.”  This statement is not correct. Surgical and laser treatments are rarely used for IH after 4-5 years of age (or any age). Please clarify.

18.  Discussion: “In our study, cases where infantile hemangiomas were in the proliferative phase and complicated or compromised vital functions were evaluated.” This sentence is confusing and should be rewritten for clarity, or deleted.

19.  Discussion: “As we are an experienced tertiary center, standard multidisciplinary professional patient follow-ups and treatments are provided for a large number of patients.” This sentence should be deleted.

Comments on the Quality of English Language

The English language is still awkward in many places, despite English editing by MDPI. Please see above for a few specific suggestions. I recommend additional editing for syntax, grammar, and sentence structure improvements.  

Author Response

We thank again  the reviewer for her/his insightful comments. 

As per the recommendations of the rewiever, detailed responses below and the corresponding revisions/corrections highlighted/in track changes in the re-submitted files.

Our work has become much more valuable thanks to you.

Round 3

Reviewer 2 Report

Comments and Suggestions for Authors

This manuscript is significantly improved from its prior versions. However, I have a few additional suggestions.

1.     Discussion: “In this study, the most commonly adverse effects of propranolol are hypotension, hypoglycemia, asymptomatic bradycardia, impaired liver function test, and hyperkalemia.”  This sentence remains incorrect and should be deleted or rewritten for clarity. In the results section, the authors specifically state that “None of the patients presented with bradycardia (< 2 SD of normal) or hypotension (< 2 SD of normal).”  Only two of 248 patients (0.8%) had hypoglycemia, and none of the patients was reported to have hyperkalemia.

2.     The next sentence says “To prevent these side effects, small babies (<8 weeks old) are hospitalized to start treatment.” Please change “prevent” to “monitor for.” Also, you should delete “small” (because “small” is not defined by a baby’s age), or replace “small babies (<8 weeks old)” with “babies <8 weeks old.”

3.     In that same paragraph, I suggest deleting “Baseline electrocardiography and echography [which should be echocardiography] are performed before treatment” because that repeats (and also contradicts) what was stated two paragraphs previously (on page 8: “In our study, all patients underwent pediatric cardiology examination with baseline electrocardiography (ECG). Echocardiography performed if indicated.”).  

Comments on the Quality of English Language

See comments to authors for additional suggestions. This manuscript is improved but still has flaws. I am still concerned that it does not add anything new or particularly noteworthy to the existing literature on infantile hemangiomas.  The English language is still awkward in several places. It may be acceptable with additional revisions (see above), and the editors are satisfied. I do not feel that I would need to see a revised version. The editors can decide next steps.  

Author Response

Thanks for your valuable comments. Once again, we thank all Reviewers and Editors for the time you put in reviewing our manuscript and look forward to meeting your expectations. We believe that the present study would provide useful information to the body of knowledge on this topic in the literature. Considering our constant efforts to improve our manuscript in terms of both scientifically and language, we kindly submitted it with relevant amendments for your consideration for publication.

Yours sincerely.

  1. Discussion: “In this study, the most commonly adverse effects of propranolol are hypotension, hypoglycemia, asymptomatic bradycardia, impaired liver function test, and hyperkalemia.”  This sentence remains incorrect and should be deleted or rewritten for clarity. In the results section, the authors specifically state that “None of the patients presented with bradycardia (< 2 SD of normal) or hypotension (< 2 SD of normal).”  Only two of 248 patients (0.8%) had hypoglycemia, and none of the patients was reported to have hyperkalemia.

ANSWER: Thanks for your valuable comments. Relevant amendments were made and the sentence was rewritten as follows:

“In this study, the most commonly adverse effects of propranolol were found to be hypoglycemia and impaired liver function tests. Hypotension, bradycardia, sleep disturbances, cold extremities, diarrhea, and gastroesophageal reflux are other reported side effects reported in the literature. ”

  1. The next sentence says “To prevent these side effects, small babies (<8 weeks old) are hospitalized to start treatment.” Please change “prevent” to “monitor for.” Also, you should delete “small” (because “small” is not defined by a baby’s age), or replace “small babies (<8 weeks old)” with “babies <8 weeks old.”

ANSWER: Thanks for your valuable comments. Based on your suggestion, the sentence was rewritten as follows:

“In order to observe these side effects, patients were hospitalized and monitored at the time of intitiation of treatment.

  1. In that same paragraph, I suggest deleting “Baseline electrocardiography and echography [which should be echocardiography] are performed before treatment” because that repeats (and also contradicts) what was stated two paragraphs previously (on page 8: “In our study, all patients underwent pediatric cardiology examination with baseline electrocardiography (ECG). Echocardiography performed if indicated.”).  

ANSWER: Thanks for your valuable comments. Based on your suggestion, “baseline echocardiography and echography are performed before treatment.” was deleted.
